# The Use of Thromboelastography in Percutaneous Coronary Intervention and Acute Coronary Syndrome in East Asia: A Systematic Literature Review

**DOI:** 10.3390/jcm11133652

**Published:** 2022-06-24

**Authors:** Ou Xu, Jan Hartmann, Yi-Da Tang, Joao Dias

**Affiliations:** 1Department of Medical Affairs, Clinical Development and Medical Safety, Haemonetics Corporation, Boston, MA 02110, USA; ou.xu@haemonetics.com (O.X.); jan.hartmann@haemoentics.com (J.H.); 2Department of Cardiology and Institute of Vascular Medicine, Peking University Third Hospital, Key Laboratory of Molecular Cardiovascular Science, Ministry of Education, Beijing 100083, China; drtangyida@126.com

**Keywords:** acute coronary syndrome, dual antiplatelet therapy, percutaneous coronary intervention, thromboelastography

## Abstract

Dual antiplatelet therapy (DAPT), alongside percutaneous coronary intervention (PCI), is central to the prevention of ischemic events following acute coronary syndrome (ACS). However, response to therapy can vary due to several factors including CYP2C19 gene variation, which shows increased prevalence in East Asian populations. DAPT responsiveness can be assessed using techniques such as light transmission aggregometry (LTA), VerifyNow^®^ and thromboelastography with the PlateletMapping^®^ assay, and there is increasing focus on the utility of platelet function testing to guide individualized treatment. This systematic literature review of one English and three Chinese language databases was conducted to evaluate the evidence for the utility of thromboelastography in ACS/PCI in East Asia. The search identified 42 articles from the English language and 71 articles from the Chinese language databases which fulfilled the pre-determined inclusion criteria, including 38 randomized controlled trials (RCTs). The identified studies explored the use of thromboelastography compared to LTA and VerifyNow in monitoring patient responsiveness to DAPT, as well as predicting ischemic risk, with some studies suggesting that thromboelastography is better able to detect low DAPT response than LTA. Other studies, including one large RCT, described the use of thromboelastography in guiding the escalation of DAPT, with some evidence suggesting that such protocols reduce ischemic events without increasing the risk of bleeding. There was also evidence suggesting that thromboelastography can be used to identify individuals with DAPT hyporesponsiveness genotypes and could potentially guide treatment by adjusting therapy in patients depending on responsiveness.

## 1. Introduction

Acute coronary syndrome (ACS) encompasses ST-segment elevation myocardial infarction (STEMI), non-STEMI and unstable angina and occurs due to a sudden blockage in the coronary arteries impeding blood flow to the myocardium [1]. Treatment is centered on restoring blood flow with the use of pharmacotherapy such as fibrinolytics, anticoagulants and antiplatelets (i.e., aspirin in combination with clopidogrel/ticagrelor; dual antiplatelet therapy [DAPT]) to dissolve thrombi and prevent restenosis, alongside non-surgical (percutaneous coronary intervention [PCI]) and surgical (coronary artery bypass graft (CABG)) interventions [2].

The monitoring of hemostasis in relation to ACS/PCI is crucial to optimize coagulation pre-PCI to reduce the risk of bleeding and thrombosis, which is usually accomplished through the use of standard laboratory tests such as platelet count and activated partial thromboplastin time, the latter of which can be used for monitoring heparinization [3,4]. In addition, platelet function tests, such as light transmission aggregometry (LTA), VerifyNow^®^ (Werfen, Warrington, UK), Multiplate^®^ (Roche Diagnostics, Rotkreuz, Switzerland) and thromboelastography (TEG^®^ analyzer; Haemonetics Corporation, Boston, MA, USA) with PlateletMapping^®^ assay can be used for assessing the degree of platelet inhibition [5,6].

The current gold standard for determining platelet reactivity, LTA, is time-consuming and requires lengthy laboratory-based sample manipulation and specialist expertise as well as large sample volumes [7,8]. Thromboelastography, VerifyNow, and Multiplate, described elsewhere [6], can be performed by non-technical staff outside the laboratory to provide rapid readouts at the patient bedside [6,9,10]. Comparison of platelet function testing using thromboelastography, Multiplate, and VerifyNow under controlled conditions has shown thromboelastography and Multiplate to have comparable results in terms of distinguishing between ticagrelor effective concentration zones, with thromboelastography showing the least variability [6]. While all three technologies assess platelet function via agonist activation of platelets, thromboelastography also takes into consideration the interaction of platelets with fibrin as well as other coagulation parameters, delivering a dynamic overview of the patient’s global hemostatic status [11].

Post-PCI platelet function testing has the potential to optimize DAPT in terms of balancing treatment response and bleeding risk (Figure 1) [12,13]. The 2018 ESC/EACTS guidelines on myocardial revascularization state that platelet function testing can be used to inform DAPT de-escalation, test treatment compliance, and provide valuable prognostic information post-PCI [14]. While the use of platelet function tests is not yet routine in clinical practice, assessment of patient response to DAPT has potential clinical value as responsiveness can differ between patients [15,16,17], and these differences may lead to differences in risk for adverse clinical events. For instance, low-on treatment platelet reactivity (LTPR) is associated with an increased bleeding risk [18], while high-on treatment platelet reactivity (HTPR) is associated with an increased risk of thrombosis and can arise due to hyper-responsiveness to therapy and hyporesponsiveness to platelet (P2Y12) antagonists such as clopidogrel, which compromises the efficacy of DAPT [18,19].

Several mechanisms can explain DAPT hyporesponsiveness such as lifestyle factors, such as smoking and obesity, as well as the use of multiple concurrent medications [15,17]. Genetic factors such as CYP2C19 gene polymorphisms also contribute to DAPT hyporesponsiveness due to ineffectual drug metabolism [16,20]. The higher prevalence of ‘CYP2C19 poor metabolizers’ in the Chinese population, where the prevalence is approximately 14% [21], suggests that platelet function testing to assess responsiveness to DAPT may be of particular interest in China as well as in other countries in East Asia. As well as genetic, there may be other factors that contribute to differing DAPT metabolism in East Asian populations, such as diet or use of traditional medicines. The adenosine diphosphate (ADP) and arachidonic acid (AA) assays enable the efficacy of P2Y12 receptor inhibitors and aspirin, used in DAPT, to be assessed, providing an indication of both hypo-responsiveness and hyper-responsiveness to therapy [22,23].

To evaluate the evidence for the utility of thromboelastography for guiding hemostatic management in ACS/PCI from East Asian centers, we performed a systematic literature review of both English and Chinese language databases.

## 2. Materials and Methods

### Literature Search

This systematic literature review was performed according to the Preferred Reporting Items for Systematic Reviews and Meta-Analyses (PRISMA) guidelines [24]. We performed a literature search in PubMed and three Chinese databases (www.cnki.net, qikan.cqvip.com and med.wanfangdata.com.cn. A search strategy (Appendix A) was used to capture publications in English or Chinese from the last 10 years. Bias was assessed using the Scottish Intercollegiate Guidelines Network (SIGN) guideline. This review reports on the findings of the literature search.

## 3. Results

### 3.1. Literature Search Results

In total, 231 articles were identified in PubMed, plus a further 451 in the 3 Chinese databases; the PRISMA flow diagram is shown in Figure 2. After screening based on the title and abstract against the inclusion/exclusion criteria (Table 1), 60 articles were identified in PubMed, and 276 articles were identified in the Chinese databases for full-text review. Of these, 153 duplicate articles were removed leaving 42 PubMed and 71 Chinese database articles that met the pre-defined inclusion criteria, which included 38 randomized controlled trials (RCTs) where patients were randomized either to different therapies or treatment algorithms based on thromboelastography use. No meta-analyses were identified (Appendix A).

The initial search strategy included terms for aortic valve replacement; however, only three studies that met the inclusion/exclusion criteria concerned this setting of care. As most of the identified literature pertained to ACS/PCI, and because treatment schedules and outcomes differ between the two settings, the focus of the current review is on the utility of thromboelastography in ACS/PCI. Identified study topics included clinical validation studies that have compared thromboelastography to the standard of care (e.g., LTA and other standard laboratory parameters), studies reporting treatment comparisons where thromboelastography was utilized and studies reporting on the utility of thromboelastography to guide DAPT use and assess hyporesponsiveness to antiplatelet therapy. Results of the SIGN bias assessment are provided in Appendix A.

### 3.2. Clinical Validation of Thromboelastography with PlateletMapping^®^ Assay

Several studies were identified that compared the use of thromboelastography vs. LTA to monitor platelet function in patients with ACS treated with PCI [8,25,26,27,28,29,30,31], with fewer studies identified comparing thromboelastography vs. VerifyNow^®^ [26,32]. A list of parameters and their associated assays discussed herein is, described in Table 2.

Tang et al., in a non-randomized prospective study of 789 patients with ACS undergoing PCI and treated with DAPT, used PlateletMapping^®^ assay percentage of platelet inhibition induced by ADP (ADP.%inhibition) to measure responsiveness to antiplatelet therapy, with thromboelastography having a strong correlation with LTA in Chinese patients (Spearman coefficient: r = 0.733, *p* < 0.001) [25]. LTA and thromboelastography were able to identify patients with major adverse cardiovascular events (MACE) at 1-year follow-up, suggesting an optimal cut-off of >53.2% for LTA and ≤32.0% for thromboelastography. LTA and thromboelastography were also able to detect HTPR associated with a three-fold risk of MACE at 1-year follow-up [25]. This large-scale prospective study provides a strong indication that thromboelastography and LTA parameters are comparable in quantifying response to DAPT and in predicting clinical outcomes, suggesting the devices could be used interchangeably.

A smaller but more recent non-randomized retrospective study (*n* = 110) compared LTA and thromboelastography for the identification of HTPR in post-PCI patients. Several PlateletMapping^®^ assay parameters were found to have a moderate correlation with LTA, including percentage of platelet aggregation rate induced by ADP and detected by thromboelastography (ADP.%aggregation), which was shown to be moderately correlated with that detected by LTA (ARADP.LTA; r = 0.5613) and was similarly indicative of HTPR [8]. Although a less strong correlation between parameters was observed in this study, the comparability of devices was supported by their similar ability to detect HTPR.

These results were in line with those of a similar-sized non-randomized prospective study of 178 patients with ACS undergoing PCI, where ADP-induced maximum platelet aggregation (ADP.MPA) and ADP-induced maximum amplitude (ADP.MA), measured by LTA and thromboelastography, respectively, were shown to have comparable positive predictive value for identifying HTPR and were predictive of patient outcomes [27]. Thromboelastography was similarly predictive of MACE in comparison with LTA; ADP.MPA and ADP.MA were significantly higher in patients with MACE (*p* < 0.05) and were independently predictive of MACE at the 6-month follow-up (ADP.MPA_,_ >46.0%, *p* = 0.001; ADP.MA, >47 mm, *p* = 0.013) [27]. Less robust data from a case control study of 425 patients also showed comparable results between LTA and thromboelastography in terms of their ability to quantify platelet aggregation rate (12% vs. 11.8%) [29]. Further evidence for the comparable performance of thromboelastography and LTA is provided by a non-randomized prospective study comparing thromboelastography and LTA after PCI in 177 patients with coronary heart disease (CHD) [31]. This study found that both methods were able to detect a high number of low DAPT responders, with more non-responders to aspirin and clopidogrel identified by thromboelastography than LTA. However, the correlation between LTA and thromboelastography was found to be poor [31].

A non-randomized prospective study looked at another platelet function test, PFA P2Y (INNOVANCE PFA-200, Siemens Healthcare GmBH, Erlangen, Germany) vs. thromboelastography and LTA in elderly cardiology patients taking clopidogrel [28]. PFA P2Y was shown to have a good correlation and co-incidence rate in comparison with LTA (r = −0.701, co-incidence rate 75%; *p* < 0.001) but a weaker correlation and co-incidence rate in comparison with thromboelastography (r = −0.475, co-incidence rate 67.9%; *p* < 0.001) [28]. Moreover, the kappa coefficient (K value) was 0.434 for PFA P2Y vs. LTA (*p* = 0.001) and 0.242 for PFA P2Y vs. thromboelastography (*p* = 0.046), where one indicates perfect agreement and zero indicates no agreement [28,33].

Overall, the identified studies that compared thromboelastography to LTA had a low risk of bias. Potential issues in some studies included small sample numbers (*n* = 110 [8]; *n* = 141, [28]), retrospective [8] and single-center design [8,25,27,28,29,31], short term follow up of cardiovascular events (1 year [25]; 6 months [27]) and a lack of randomization [8,25,27,28,31,34].

Fewer studies were identified that compared thromboelastograhy to VerifyNow^®^. Comparison of VerifyNow^®^ and thromboelastography in patients with CHD undergoing PCI in combination with DAPT (aspirin and clopidogrel) in 184 Chinese patients showed that VerifyNow^®^ was comparable to thromboelastography for on-clopidogrel platelet reactivity (r = −0.511) [32]. Identification of HTPR using VerifyNow^®^ showed significant but poor agreement with thromboelastography (K = 0.225). A significant moderate agreement was found for LTPR (κ = 0.412) [32]. This relatively small study suggests that thromboelastography shows comparable performance to VerifyNow^®^ in identifying platelet reactivity in patients receiving DAPT, but there were differences in quantification of HTPR and LTPR.

A further study identified suggested that the performance of VerifyNow^®^ and thromboelastography is comparable. This study, by Koh et al., compared the utility of the three methodologies: VerifyNow^®^, LTA and thromboelastography, to assess the antiplatelet effects of a fixed-dose combination (FDC) capsule (HCP0911) of DAPT combining clopidogrel and aspirin in patients treated with coronary stents. Each method showed comparable results demonstrating that the pharmacodynamic effect of HCP0911 was non-inferior to the separate administration of clopidogrel and aspirin [26].

Studies that included comparison of VerifyNow^®^ to thromboelastography mostly had a high methodological quality. Potential issues again included small sample size (*n* = 184, [32]; *n* = 30, [26]), single-center design [26,32] and lack of randomization [32]. Of note, Koh et al. examined only male Korean patients and assessed patient compliance using a questionnaire as opposed to testing [26].

A further paper identified in the literature search was a Chinese consensus statement that provides recommendations for the use of various platelet activation tests and concludes that thromboelastography is one of the reliable techniques in the detection of platelet function [30]. Our review of the literature from Chinese/East Asian centers supports this, with the largest study identified [25] showing that thromboelastography parameters correlate strongly with those of the current gold standard assay (LTA) and that thromboelastography can predict MACE/ischemic risk in a similar manner to LTA. This was supported by other studies that showed comparable performance between LTA and thromboelastography in detecting platelet reactivity and/or predicting clinical outcomes, although one smaller study [31] showed a poor correlation between the assays. Findings in relation to the performance of thromboelastography vs. LTA from East Asian Centers, as summarized in Table 3, suggest that thromboelastography has comparable performance vs. the current gold standard method (LTA) and could be considered as an alternative to LTA. Fewer studies were identified that compared thromboelastography to VerifyNow^®^; the limited data identified suggests that they are comparable, but there may be some differences, particularly with respect to the identification of HTPR.

### 3.3. Use of Thromboelastography to Quantify Antiplatelet Efficacy in Therapy Comparison Studies

Several studies from East Asian centers have shown that thromboelastography platelet function parameters are affected by different treatment regimens (Figure 3) [35,36,37,38]. In one single-center RCT, 120 patients undergoing emergency PCI following an acute myocardial infarction (AMI) were perioperatively assessed following infusion with either ticagrelor or clopidogrel [35]. Measurement of thromboelastography parameters showed that ticagrelor can improve antiplatelet response in patients with AMI during the perioperative period of emergency PCI without increasing bleeding risk as shown by lower parameters including α-angle, CK.MA [citrated kaolin], ADP.MA and AA.MA (*p* < 0.05) [35]. In a further RCT of 120 acute STEMI patients after emergency PCI, ADP.MA determined by thromboelastography was used to investigate the effects of different doses of atorvastatin on plasma endothelin and platelet function [36]. The ADP.MA in the group receiving 40 mg of atorvastatin (intensive group, *n* = 60) was shown to be significantly lower than the group receiving 20 mg of atorvastatin (standard group, *n* = 60) after treatment: 38.4 ± 17.4 mm vs. 45.7 ± 14.5 mm, respectively, *p* < 0.05. The authors conclude that high intensity atorvastatin therapy during PCI reduces plasma endothelin, improves endothelial function and reduces residual platelet activity [36].

Studies examining ACS patients treated with tirofiban have utilized thromboelastography to assess on-treatment platelet reactivity [37,38]. A non-randomized prospective cohort study by Li et al. (2019) investigated platelet function and the risk of bleeding in 196 patients with ACS receiving treatment with (*n* = 98, treatment group) or without (*n* = 98, control) tirofiban. Thromboelastography was used to assess platelet inhibition following stimulation with AA and ADP, with platelet inhibition being significantly higher in patients treated with tirofiban compared to the control group: 80.3% ± 19.6% vs. 72.6% ± 13.0% (*p* = 0.002) and 81.0% ± 19.8% vs. 75.4% ± 12.4% (*p* = 0.020), respectively [37]. Another single-center RCT compared the use of double (aspirin and clopidogrel) or triple (aspirin, clopidogrel and tirofiban) antiplatelet therapy in 150 STEMI patients post-PCI, showing the AA and ADP inhibition rate to be significantly higher in the triple group compared to the double group (*p* < 0.01). Additionally, the ADP.MA of the triple group was significantly lower than the double group (*p* < 0.01) [38].

In general, identified studies for the use of thromboelastography in antiplatelet therapy were in the SIGN risk-of-bias evidence Grades 1 and 2. Potential issues were small sample size (*n* = 120, [35]; *n* = 120, [36]; *n* = 196, [37]), single-center studies [35,37,38], lack of randomization [37] and lack of follow-up [38]. A potential source of bias in the cohort study by Li et al. (2019) was that patients with culprit vessel stenosis and multi-coronary lesions tended to receive tirofiban treatment as a matter of course [37].

Overall, the studies identified show that thromboelastography can detect hemostatic changes resulting from the escalation of therapy as outlined in Figure 3. The capacity of thromboelastography to detect these changes underpins its potential to guide individualized treatment of DAPT. 

### 3.4. Utility of Thromboelastography to Monitor Response to Therapy and for Individualized Treatment

Several studies from East Asia used thromboelastography to monitor the effectiveness of antiplatelet therapies and assess thrombotic risk [34,39,40,41,42]. This could potentially be used to guide individualized treatment by adjusting therapy depending on patient response. 

In a study by Zhong et al., investigators aimed to assess the performance of thromboelastography in monitoring coagulation status and antiplatelet efficacy in 71 CHD patients, with comparison to a cohort of 380 healthy individuals with normal thromboelastography parameters [39]. In this single-center RCT, thromboelastography parameters (ADP.MA, CK.MA and act.MA [activator]) were shown to be indicative of thrombogenesis risk: 9.86%, 4.23% and 12.68%, respectively [39]. The authors conclude that thromboelastography can guide and individualize DAPT for CHD patients, improving the safety of anti-thrombotic therapy [39].

Another prospective single-center study in 447 ACS patients found ADP.MA and AA.MA to be indicative of net residual platelet reactivity and predictive of 6-month ischemic risk after aspirin or clopidogrel treatment [40]. These platelet reactivity measures, ADP.MA and AA.MA, were shown to be superior to platelet inhibition rate such as AA.%inhibition and ADP.%inhibition in assessing thrombotic risk. The authors conclude that thromboelastography may be used to provide individualized prognostic information on the efficacy of antiplatelet drugs as well as the risk of thrombosis [40].

Other studies have used thromboelastography to identify patients insensitive to antiplatelet drugs to develop an individualized therapy [34,41]. In a retrospective study by Wu et al., (2016), 168 patients with CHD were divided into three groups: combined drug group (*n* = 56, aspirin and clopidogrel; PCI treatment), aspirin group (*n* = 56, aspirin only; no PCI treatment) and clopidogrel group (*n* = 56; no PCI treatment). Thromboelastography showed the platelet AA inhibition rate of the combined drug group was comparable to that of the aspirin group (*p* = 0.05), and the ADP inhibition rate was comparable to that of clopidogrel group [41].

Another non-randomized prospective study of 300 patients with CHD post-PCI showed thromboelastography to detect more low responders than LTA [34]. Thromboelastography had a detection rate of 29.00% for aspirin low responders and 31.00% for clopidogrel low responders when compared to LTA, which had a detection rate of 18.67% for aspirin low responders and 23.00% for clopidogrel low responders (*p* < 0.05). Patients who were insensitive to antiplatelet therapies appeared to have a greater risk of MACE (7.14% risk in low responders vs. 3.96% in sensitive patients; *p* > 0.05), with the authors concluding that the detection of additional low responders using thromboelastography was beneficial for adjusting DAPT and improving clinical outcomes [34].

A retrospective case control study showed that ischemic and bleeding events were reduced in the 2 months after surgery for patients treated with DAPT and monitored using thromboelastography (*n* = 206) vs. those whose platelet function was not monitored (*n* = 206) [42]. The inhibitory rate of aspirin and clopidogrel prior to thromboelastography was 87.26 ± 23.15% and 60.24 ± 30.37%, respectively. Following DAPT adjustment guided by thromboelastography, the inhibitory rate of aspirin and clopidogrel was 95.72 ± 12.74% and 71.33 ± 22.58%, respectively. The probability of adverse cardiovascular ischemia and bleeding events was 7.77% and 2.42%, respectively, in the thromboelastography group vs. 21.84% and 5.34% in the control group (*p* < 0.05) [42].

Some potential issues include the lack of large-scale randomized control trials in this area, with most studies being single-center [34,39,40,41,42], cohort [34,41], observational [40] and retrospective [40,41,42] design, with a lack of randomization [34,42] and a lack of follow-up [41,42]. In the studies by Wu et al. (2016) and Tái et al., the use of inclusion and exclusion criteria is unclear [41,42]. Moreover, the adjustment of the antiplatelet protocol according to thromboelastography was not clear in the study by Tái et al. [42].

In summary, the identified studies suggest that thromboelastography can identify non-responders and guide DAPT adjustment, individualizing therapy and improving antiplatelet efficacy. There was also some evidence to suggest that thromboelastography-guided optimization of DAPT reduced the risk of MACE and bleeding events.

### 3.5. Utility of Thromboelastography for Guiding Escalation of Antiplatelet Therapy

Several of the identified studies from East Asia also assessed the utility of thromboelastography in specifically guiding therapy escalation and assessing the effect of different genetic polymorphisms on platelet responsiveness [16,43,44,45,46]. The CREATIVE trial (Clopidogrel Response Evaluation and Anti-Platelet Intervention in High Thrombotic Risk PCI Patients) assessed the safety and efficacy of intensive antiplatelet therapy vs. standard therapy in PCI patients. This large RCT utilized thromboelastography to identify 1078 PCI patients at a greater risk of thrombosis (ADP.MA > 47 mm; platelet inhibition rate < 50%). Patients were assigned to receive either standard therapy (aspirin plus clopidogrel 75 mg [*n* = 362]), aspirin plus clopidogrel 150 mg (*n* = 359) or aspirin plus clopidogrel 75 mg and cilostazol (*n* = 355). Over an 18-month follow-up period, the incidence of the primary endpoint (MACE or cerebrovascular events) was lower in the double-dose clopidogrel (10.6%) and the triple therapy groups (8.5%) than the standard therapy group (14.4%; hazard ratios [95% confidence intervals]: 0.720 [0.474–1.094] and 0.550 [0.349–0.866] for double and triple therapy vs. standard therapy, respectively). The risk of major bleeding in the escalated therapy groups vs. the standard therapy groups was not significantly different; the authors, therefore, conclude that thromboelastography-guided escalation of therapy can improve outcomes in patients with low responsiveness to clopidogrel [45]. Moreover, genotype testing showed >60% of patients to have CYP2C19 gene polymorphisms with extensive, intermediate, and poor CYP2C19 metabolizing capability [45].

A non-randomized prospective study of 168 patients with coronary artery disease investigated the correlation between CYP2C19 and ABCB1 and ischemic events [43]. Patients with a platelet inhibition rate of <30%, assessed by thromboelastography, were classified as having high platelet responsiveness (HPR, *n* = 50) and compared with a normal platelet responsiveness (NPR) group (*n* = 118). CYP2C19*3 incidence was significantly higher in the HPR vs. the normal group (81.82% vs. 18.18%; *p* < 0.001). The incidence of CYP2C19*2 and ABCB1 3435CT was also higher in the HPR vs. the normal group; however, this was not significant (*p* = 0.234 and *p* = 0.157, respectively) [43].

A non-randomized case–control study of 459 ACS patients receiving clopidogrel and aspirin also used thromboelastography in this way. Patients with <30% platelet inhibition were identified as HPR and compared with NPR patients. Genotype distribution was examined for both groups. Thromboelastography was able to distinguish clopidogrel hyporesponsiveness variants based on the significant influence of the CYP2C19 and PON1 Q192R variants on ADP-induced platelet inhibition [16].

On the other hand, another case control study of 124 patients with CHD treated with PCI found no correlations between thromboelastography platelet function parameters and the CYP2C19 genotype [44]. Patients were divided into three groups (normal, intermediate and slow clopidogrel metabolism) according to CYP2C19 genotype. Thromboelastography was used to assess ADP.%inhibition and ADP.MA. However, the incidence of ADP.MA > 47 mm across the three groups was not statistically significant (χ^2^ = 1.883, *p* > 0.05) [44].

In a further study, CYP2C19 genotyping in combination with thromboelastography identified clopidogrel hyporesponsiveness and enabled therapy escalation with ticagrelor [46]. In this single-center cohort study of 124 subjects, patients with CHD post-PCI were tested for CYP2C19 gene polymorphisms and divided into two groups: normal and hyperbolic types and abnormal metabolic (including intermediate and slow metabolic). Patients were also assessed using thromboelastography and divided into groups based on response to clopidogrel: clopidogrel normal reaction type (NCR, platelet inhibition rate ≥ 30%) and clopidogrel low reactive type (LCR, platelet inhibition rate < 30%). Patients with normal metabolism who were NCR or LCR, or those who had abnormal metabolism and were NCR, were given aspirin and clopidogrel. Patients with abnormal metabolism that were LCR were given aspirin and ticagrelor. Ticagrelor was shown to be more effective reducing restenosis in clopidogrel non-responders and CYP2C19 mutation cases without increasing major bleeding events [46].

Identified studies in relation to the utility of thromboelastography in identifying hyporesponsiveness to antiplatelet therapy and in guiding escalation of antiplatelet therapy were deemed to be generally of low risk of bias. Potential issues were lack of randomization [16,43,44,46], single-center design [16,43,44,45,46], short or unclear follow-up [16,43,46] and small sample numbers (*n* = 459, [16]; *n* = 168, [43]; *n* = 124, [44]; *n* = 124, [46]). The authors of Peng et al., noted that no HPR risk assessment was possible for CYP2C19∗3/∗3 and CYP2C19∗17/∗17, citing sample size as the limiting factor [16]. Moreover, this paper was an observational case control study and was consequently subject to inherent risks of selection bias [16]. The authors of Tang et al. also discussed a limitation which may introduce bias relating to the frequency of early discontinuation of the study drug in the double and triple treatment strategies vs. the standard strategy [45].

Overall, these studies suggest that thromboelastography has potential utility for guiding individualized treatment, particularly in terms of escalating therapy and improving efficacy of DAPT. Thromboelastography was shown to effectively identify clopidogrel hyporesponsiveness and guide treatment escalation with cilostazol or ticagrelor to reduce adverse events and improve patient outcomes.

## 4. Discussion

This systematic literature review determined that thromboelastography has been extensively used in East Asia (principally China) in relation to ACS and PCI, and studies from this region support the correlation of thromboelastography results with traditional laboratory parameters, i.e., LTA, as well as other whole blood near-patient platelet function testing assays (VerifyNow^®^) despite the differences in readouts from these assays. In a variety of prospective studies, thromboelastrography platelet function parameters showed associations with cardiovascular outcomes, and there was some evidence to suggest that thromboelastography can guide individualized DAPT treatment. 

In the cardiology setting, thromboelastography is a well-established tool in cardiac surgery; thromboelastography-based transfusion algorithms are widely used to reduce bleeding and transfusion requirements [47,48]. To our knowledge, this is the first systematic review of the available East Asian literature on the use of thromboelastography in relation to DAPT/PCI in interventional cardiology. Results of this literature review are consistent with observations from a Western perspective supporting a role of thromboelastography in relation to PCI and personalized DAPT treatment [49].

Thromboelastography can provide rapid near-patient assessment of platelet function and provide clinicians with a valuable assessment of patients’ response to anti-platelet therapy, and may, therefore, be a good candidate to guide clinical decision-making [22]. The most recent TEG^®^ device that can run the PlateletMapping^®^ function is the TEG^®^6s. Although most of the data reported in the present study is based on the older TEG^®^5000, readings from the TEG^®^5000 andTEG^®^6s devices have been shown to be well correlated when used in the settings of cardiology (including intra-individual reproducibility) [50], cardiac surgery [51] and trauma [52]. The present review identified several studies from East Asian centers that cited similar performance of thromboelastography to the gold-standard assay (LTA) in terms of identifying HTPR and predicting clinical outcomes, with the largest study identified in this regard [25] showing a strong correlation between thromboelastography and LTA parameters. This is in line with observations from Western centers [22,53,54]. Since thromboelastography has been found to correlate with the gold-standard LTA in both Western and East Asian populations, and because the results of thromboelastography are available rapidly and testing can be performed near the patient, thromboelastography may be a more viable method to tailor DAPT treatment than traditional laboratory testing via LTA. However, although MA.ADP appears to reliably predict long-term outcomes, there is a lack of clarity as to whether it is the parameter of greatest utility for guiding treatment. The thromboelastography parameters investigated varied between studies, and several different parameters (e.g., ADP.MA and ADP.%inhibition/%aggregation) identified HTPR and predicted clinical outcomes. Future RCTs should, therefore, seek to identify the optimal thromboelastography parameter for guiding DAPT treatment. 

Currently, recommendations for the use of near-patient assessments, such as thromboelastography, to tailor antiplatelet use vary in guidelines [12,55,56], with recent European guidelines suggesting that whole blood platelet function testing may be used to guide de-escalation of therapy but not escalation [55]. The ESC 2017 guidelines state that “neither platelet function testing nor genetic testing can be recommended for tailoring DAPT. It may be considered in specific situations (e.g., patients suffering from recurrent adverse events) if the results may change the treatment strategy. This is the case for patients undergoing CABG who are exposed to DAPT” [55]. An international expert consensus statement (with key leaders from Asia as well as North America and Europe) states that “In selective scenarios, PFT and genotyping may be used as optional tools for guiding treatment” [12]. The 2020 Asian Pacific Society of Cardiology Consensus Recommendations currently do not recommend the use of platelet function guided therapy due to a lack of prospective randomized trials in the Asia–Pacific region [56]. The present review identified more specific evidence from studies in East Asia for the utility for thromboelastography to guide escalation of therapy, i.e., in relation to therapy hyporesponsiveness, which may reflect differing research priorities in East Asian centers than in Western centers. The most robust study identified in this regard was the CREATIVE trial; this RCT provides strong evidence that escalation of DAPT based on a low thromboelastography-determined platelet inhibition rate improves clinical outcomes over an 18-month follow-up period without compromising safety [45]. Nevertheless, there is an overall need for further evidence on this topic; in particular, the optimal timing of thromboelastography monitoring of DAPT is unknown.

As most studies on clopidogrel responsiveness have been performed in Caucasian populations, there is a need for more clinical studies in more diverse populations to take into consideration factors such as diet and lifestyle [57]. Some studies in East Asian patients have shown the effect of P2Y12 inhibition to vary, a phenomenon referred to as the East Asian Paradox, whereby patients of East Asian descent exhibit differing thrombotic and bleeding risks when treated with different P2Y12 inhibitors [58]. The higher prevalence of the CYP2C19 genotype in East Asian individuals is likely a significant contributor to this. Future comparison studies may be warranted to address the population differences based on genotype or compare outcomes based on background, as this study did not compare thromboelastography results or effectiveness between East Asian and Western populations.

Results of this literature review suggest that thromboelastography may be a more efficient way of assessing whether patients will respond to clopidogrel compared with genetic testing, which has associated time and cost implications and the availability of which is likely highly variable. Not all studies found correlations between genetic variants and thromboelastography parameters. However, despite this, thromboelastography was shown to detect low responsiveness to clopidogrel, and as other factors influence responsiveness to clopidogrel therapy, thromboelastography may provide a more precise indication of therapy requirements vs. genotyping, thus demonstrating the value of a functional assay alongside—and potentially in place of—genotyping [59]. Moreover, thromboelastography has the additional benefit of providing a global overview of hemostasis [60], helping to guide the need for therapy escalation as well as identifying any coagulation defects that could contribute to poor outcomes in ACS/PCI patients.

Some limitations to the present review should be noted. Chinese articles were assessed by one author only, whereas English language articles were assessed by two independent assessors. Nevertheless, a systematic approach was applied to decrease bias. To maximize the available data for this review, data from a range of study types were considered and included. However, as a result, there was variation across studies in terms of experimental design. Caution should, therefore, be used when comparing between studies.

## 5. Conclusions 

This systematic literature review demonstrates that thromboelastography has been widely used and validated in East Asia for monitoring hemostasis in relation to ACS/PCI and DAPT and that thromboelastography platelet activity parameters are associated with cardiovascular risk. Several studies provide observations consistent with those from Western centers that thromboelastography has the potential to guide individualized antiplatelet therapy. Genetic hyporesponsiveness to DAPT appears to be a particular concern for East Asian clinicians, and thromboelastography has shown utility here in guiding antiplatelet management.

## Figures and Tables

**Figure 1 jcm-11-03652-f001:**
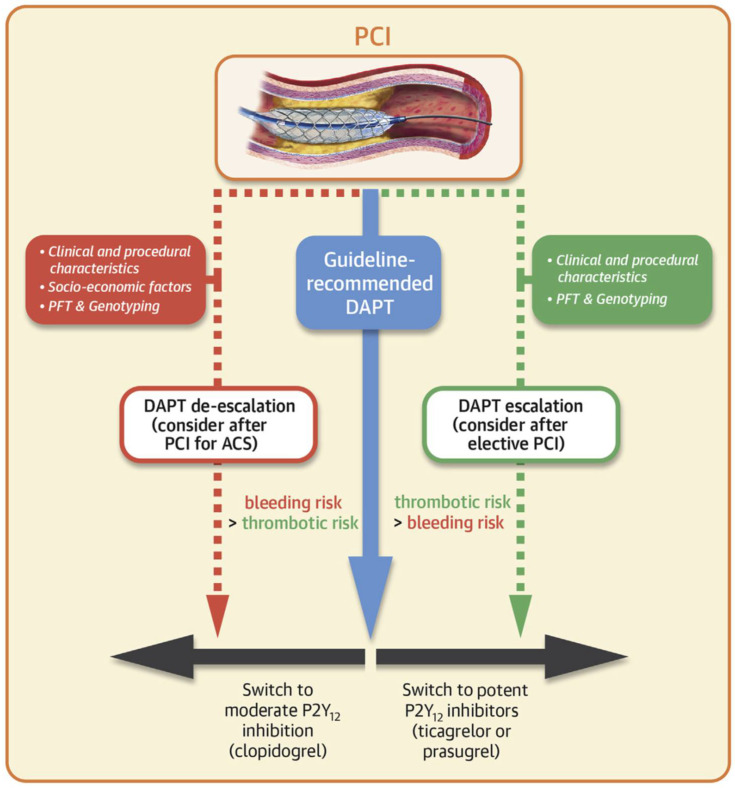
Considerations when implementing DAPT in relation to PCI; PFTs (such as thromboelastography) can guide escalation/de-escalation of therapy. Reproduced with permission from Sibbing D. et al., *JACC Cardiovasc Interv* 2019 [12]. ACS: acute coronary syndrome; DAPT: dual antiplatelet therapy; PCI: percutaneous coronary intervention; PFT: platelet function tests.

**Figure 2 jcm-11-03652-f002:**
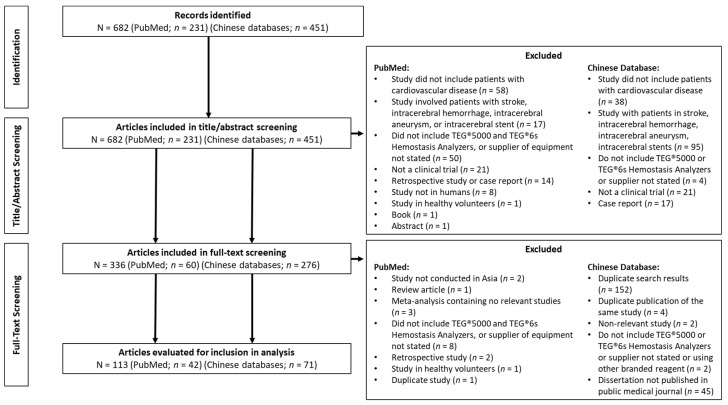
PRISMA diagram. TEG: thromboelastography.

**Figure 3 jcm-11-03652-f003:**
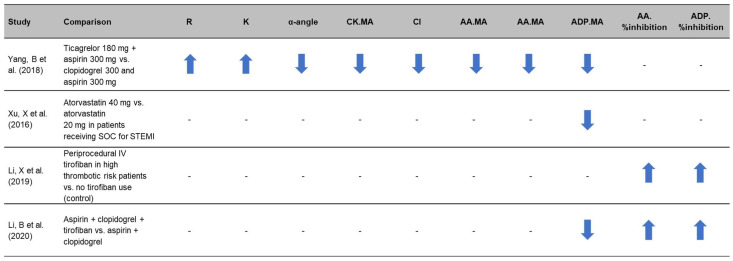
Thromboelastography parameters are impacted by therapy escalation, reflecting greater platelet inhibition [35,36,37,38]. Arrows reflect increase (up) or decrease (down) in the parameters. AA: arachidonic acid; ADP: adenosine diphosphate; CK: citrated kaolin; CI: coagulation index; K: kinetics; MA: maximum amplitude; R: reaction time; SOC: standard of care; STEMI: ST-segment elevation myocardial infarction.

**Table 1 jcm-11-03652-t001:** Inclusion and exclusion criteria.

Inclusion Criteria	Exclusion Criteria
English language article OR Chinese language article	Any language other than English or Chinese
Studies of humans or human blood samples (adults or pediatrics)	Non-human studies, any studies performed in animals
Authors/investigators from any center involved are from an Asian country	Authors/investigators from a country not in Asia
Clinical trials and meta-analyses	Reviews, case reports, editorials, responses, comments, congress abstracts
Studies with prospective data collection	Studies based on retrospective data collection
Reports data relevant to cardiology	Reports data from a setting other than cardiology
Utilization of standard TEG (5000 and 6s) in the context of predicting or improving patient outcomes	No Viscoelastic testing data reported Viscoelastic data reported but not directly linked to assessment/treatment of cardiovascular disease Use of other VHA device (e.g., ROTEM, SONOCLOT, Multiplate^®^, VerifyNow^®^, Haema system, CFMS LEPU) only Information about the supplier of equipment not provided

VHA: viscoelastic hemostatic assay; TEG: thromboelastography; ROTEM: rotational thromboelastometry.

**Table 2 jcm-11-03652-t002:** Descriptions of platelet function/hemostatic parameters.

Assay	Parameter	Description
Thromboelastography	α-angle	Rate of clot formation
	CK.MA	Measures the fibrin formation phase; overall clot strength (mostly driven by platelet count and function as well as fibrin formation) and stability showing platelet and fibrin interacting via GPIIb/IIIa
	act.MA	Assesses clot strength without platelet contribution
	ADP.MA	Functional component of platelet clot strength derived by ADP-agonist stimulation (for pharmacologic inhibition of ADP pathway using anti-P2Y12 therapies, i.e., clopidogrel, ticagrelor, prasugrel)
	AA.MA	Functional component of platelet clot strength derived by AA-agonist stimulation (for pharmacologic inhibition of AA pathway using thromboxane pathway blockers, i.e., aspirin)
	ADP.%aggregation	Percentage platelet aggregation rate induced by ADP (calculated from ADP.MA−Fibrin.MA/Thrombin.MA−Fibrin.MA × 100)
	AA.%aggregation	Percentage platelet aggregation rate induced by AA (calculated from AA.MA−Fibrin.MA/Thrombin.MA−Fibrin.MA × 100)
	ADP.%inhibition	Percentage clot strength change due to platelet function inhibition induced by ADP (calculated from platelet aggregation: [(ADP.MA−Fibrin.MA_n_)/(Thrombin.MA−Fibrin.MA) × 100] and %inhibition: [100% platelet aggregation])
	AA.%inhibition	Percentage clot strength change due to platelet function inhibition induced by AA (calculated from platelet aggregation: [(AA.MA−Fibrin.MA_n_)/(Thrombin.MA−Fibrin.MA) × 100] and %inhibition: [100% platelet aggregation])
LTA	ADP.MPA	ADP-induced maximum platelet aggregation
	ARADP.LTA	ADP-induced aggregation rate by LTA

AA: arachidonic acid; act: activator F; ADP: adenosine diphosphate; CK: citrated kaolin; GPIIb/IIIa: glycoprotein IIb/IIIa; LTA: light transmission aggregometry; MA: maximum amplitude.

**Table 3 jcm-11-03652-t003:** Performance of thromboelastography with PlateletMapping^®^ assay vs. the current gold-standard assay in studies in China (LTA) [8,25,27,29,31].

Study	Drug Intake and Timing of Assay Utilization	TEG^®^ Device Used	Between-Parameter Correlations	Identification of HTPR	Prediction of MACE/Ischemic Risk		Summary
					Thromboelastography	LTA	
Tang, X F et al. (2015) [24] N = 789	Loading doses (12 h prior to PCI)—therapy-naïve patients: 300 mg DAPT; patients previously on antiplatelet therapy: 100 mg aspirin, 75 mg clopidogrel Daily maintenance dose following surgery: 100 mg aspirin, 75 mg clopidogrel TEG^®^ assay carried out 6 h after clopidogrel dose	TEG^®^5000	Spearman coefficient for ADP.%inhibition vs. ARADP.LTA: r = 0.733, *p* < 0.001	HTPR cutoff for ADP.%inhibition (≤32%) found in 36.1% of enrolled subjects HTPR cutoff for ARADP.LTA: (53.2%) found in 29% of subjects	ROC curve analysis AUC, % (95% CI) = 0.684 (0.650–0.716), 0.0001 1-year MACE occurred in 6.7% with and 2.6% without HTPR	ROC curve analysis AUC, % (95% CI) = 0.677 (0.643–0.710), *p* = 0.0009 1-year MACE occurred in 7.4% and 2.7% without HTPR	Thromboelastography has shown strong performance for detecting low DAPT response/HTPR, with a high sensitivity and specificity for detecting HTPR (similar to LTA); Thromboelastography has also shown comparable performance to LTA in predicting ischemic risk at 6 months and 1 year; The strength of correlations between thromboelastography and LTA parameters varied; the strongest correlation was reported in the largest study and between ADP.%inhibition and ARADP.LTA
Cheng, D et al. (2020) [7] N = 110	Loading dose: aspirin 75 mg, ticagrelor 180 mg Maintenance dose: aspirin 75 mg, ticagrelor 90 mg TEG^®^ and LTA assays were ordered simultaneously (no specific time given)	TEG^®^5000	ADP.%aggregation vs. ARADP.LTA: r = 0.5613, *p* < 0.01 ADP.MA vs. ARADP.LTA: r = 0.5567, *p* < 0.01 Net ADP.MA vs. ARADP.LTA: r = 0.5836, *p* < 0.01	AUCs (95% CI) for ROC curve analysis: * *ADP.%aggregation (%)* 0.8199 (0.734–0.886); cutoff = 64.6; sensitivity = 82.61; specificity = 80.46 *ADP.MA (mm)* 0.812 (0.726–0.880); cutoff = 45.6, sensitivity = 78.26; specificity = 81.61 *Net ADP.MA (mm)* 0.849 (0.768–0.910); cutoff = 26.3; sensitivity = 91.30; specificity = 73.56	-		
Tang, N et al. (2015) [26] N = 178	Loading dose (prior to PCI)—therapy-naïve patients: 300 mg clopidogrel; patients previously on antiplatelet therapy: 75 mg clopidogrel Daily maintenance dose following surgery: 100 mg aspirin, 75 mg clopidogrel Blood samples collected 18–24 h post-PCI.	TEG^®^5000	-	*ADP.MA* HTPR defined as >47 mm; positive predictive value = 31.6%, negative predictive value = 91.7% *ADP.MPA* HTPR defined as >46%; positive predictive value = 33.3%, negative predictive value 97.6%	ADP.MA in patients with MACE vs. those without: 43.5 ± 20.6% vs. 33.0 ± 15.2, *p* = 0.021 ADP.MA > 47 mm independently predicted 6-month MACE (*p* = 0.013)	MPA.MA in patients with MACE vs. those without: 52.9 ± 19.2% vs. 29.4 ± 18.7%, *p* = 0.002 MPA.MA > 46% independently predicted 6-month MACE (*p* = 0.001)	
Li, G et al. (2017) [28] N = 425	DAPT: aspirin 100 mg/day, clopidogrel 75 mg/day Blood sample taken 3 days after treatment start	TEG^®^5000	ADP.%aggregation (11.8%) vs. ARADP.LTA (12.0%): r = 0.351, *p* = 0.01	-	-	-	
Miao, L et al. (2017) [30] N = 177	Loading dose for therapy-naïve patients: 300 mg DAPT Daily maintenance dose: 100 mg aspirin, 75 mg clopidogrel Blood sample for TEG^®^ and LTA assay taken one month after PCI	TEG^®^5000	Weak correlations between TEG and LTA	Detection rates of low DAPT response: LTA = 30.3%; thromobelastography = 45.5%	-	-	

* Using definition of HTPR as ARADP.LTA >46%; ADP: adenosine diphosphate; ARADP: ADP-induced aggregation rate; AUC: area under the curve; CI: confidence interval; DAPT: dual antiplatelet therapy; HTPR: high on-treatment platelet reactivity; LTA: light transmission aggregometry; MA: maximum amplitude; MPA.MA: ADP-induced maximum platelet aggregation; ROC: receiver operative characteristic. Tang XF 2015, Cheng D 2020 and Tang N 2015 were identified from PubMed, while Li 2017 and Miao 2017 were from the Chinese database.

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
