# Peer review of "The Use of Thromboelastography in Percutaneous Coronary Intervention and Acute Coronary Syndrome in East Asia: A Systematic Literature Review"

_jcm, 2022, doi:10.3390/jcm11133652_

Round 1
Reviewer 1 Report
The response to the dual antiplatelet therapy (DAPT), alongside percutaneous coronary intervention (PCI) may be variable due to several factors including CYP2C19 gene variation, which shows increased prevalence in East Asian populations.
The authors performed systematic literature review of English and Chinese language databases to evaluate the evidence for the utility of thromboelastography in ACS/PCI in East Asia.
The article is comprehensive and it fulfills all the criteria of the systematic review.
It is written on seventeen pages, two of which provide references. The results of the study are properly documented by the use of three figures and five tables. Materials and methods are described transparently.
On the basis of the results of the present study the authors came to these conclusions:
- several studies were identified suggesting that thromboelastography was comparable to LTA and VerifyNow in monitoring patient responsiveness to DAPT, as well as predicting ischemic risk, with some studies suggesting that thromboelastography is better able to detect low DAPT response than LTA.
- other studies described the use of thromboelastography in guiding escalation of DAPT, with some evidence suggesting that such protocols reduce ischemic events without increasing the risk of bleeding.
- there was also evidence suggesting that thromboelastography can be used to identify individuals with DAPT resistance genotypes and can guide treatment in these patients.
Author Response
The response to the dual antiplatelet therapy (DAPT), alongside percutaneous coronary intervention (PCI) may be variable due to several factors including CYP2C19 gene variation, which shows increased prevalence in East Asian populations.
The authors performed systematic literature review of English and Chinese language databases to evaluate the evidence for the utility of thromboelastography in ACS/PCI in East Asia.
The article is comprehensive and it fulfills all the criteria of the systematic review.
It is written on seventeen pages, two of which provide references. The results of the study are properly documented by the use of three figures and five tables. Materials and methods are described transparently.
On the basis of the results of the present study the authors came to these conclusions:
- several studies were identified suggesting that thromboelastography was comparable to LTA and VerifyNow in monitoring patient responsiveness to DAPT, as well as predicting ischemic risk, with some studies suggesting that thromboelastography is better able to detect low DAPT response than LTA.
- other studies described the use of thromboelastography in guiding escalation of DAPT, with some evidence suggesting that such protocols reduce ischemic events without increasing the risk of bleeding.
- there was also evidence suggesting that thromboelastography can be used to identify individuals with DAPT resistance genotypes and can guide treatment in these patients.
Response: We thank the reviewer for their comments and their time to review our manuscript
Reviewer 2 Report
This is a systematic review in the field of (D)APT, that deals with two significant issues
- potential ethnical differences in the antiplatelet effects
- potential usefulness of VET / TEG
The issue of escalation and de-escalation is important indeed (see for instance
doi: 10.1016/j.amjcard.2020.12.020.PMID: 33706987)
Regarding the first point, the authors should better delineate the clinical impact: Are ethnic differences potentially clinically relevant? How many patients of East-Asian ascent were included in the published clinical studies? North-Am based studies vs. Asian studies. What were the outcomes according to ethnical background? Might traditional Chinese medicines if used ‘on top of’ APT have played a role? Were their uses controlled for? And what about dietary influences?
Is there any interventional cardiologist working in China among the authors? Author Contributions: J.H., J.D. and O.X. contributed to the design of the study; J.D. and O.X. con- 
L.251 -tributed to the screening of papers and data extraction; all authors contributed to the interpretation 
L.252 of the data, writing of the manuscript and reviewed/approved the final version for submission. 
What was the role of Tang Yi-Da then?
L.256 Conflicts of Interest: J.H., J.D. and O.X. are employees of Haemonetics. The remaining authors de- 
L.257 -clare no competing interests: just one remaining author! Namely Tang Yi-Da.
Importantly, the authors conclude LL173 et sq. that “studies 
from this region support the comparability of TEG to traditional laboratory parameters, i.e., LTA, as well as other whole blood near-patient platelet function testing assays (VerifyNow® ), in the context of PCI/AMI. Studies from East Asia have also 
hown TEG to be effective in guiding hemostasis management in relation to individualized treatment, and that TEG platelet FUNCTION parameters are associated with cardiovascular outcomes”. This is not fully supported by the literature: first one, to some extent only – PFT using different read-outs can’t be easily compared – classification of patients according to the extent of platelet inhibition; second: I strongly disagree - just one single RCT?; third: I agree – but should be in the second place.
Literature search and grading were well conducted and described. Why was a metaanalysis not possible? Of note, English but also Chinese language databases were included. For the latter, just one author could analyse them, which is a study limitation acknowledged by the authors.
PRISMA guidelines were used. 
Tables are welcome: S1: Search terms; S2: SIGN grading 
but Table S2 should mention the language of the retrieved studies. I have doubts about the evaluation of methodological quality of identified studies. For instance P.13 it reads “…deemed to be generally of 
L.156 low risk of bias. »â€¨ The authors do not provide convincing evidence for such an appreciation.
Studies in the setting of AVR were few and the authors decided not to incorporate them into their review: please tell us how many they were.
The retrieved studies are with TEG only by design of the systematic review; hence ‘thromboelastography’ should be ‘viscoelastometric testing (VET) with TEG®’. Admittedly TEG with PM (PlateletMapping®) is the most elaborated approach to study with VET the effect of APT on platelets – it is a complicated one, but the authors could have discussed why other VET devices with the use of an inhibitor of the platelet component and platelets stimulated by endogenous thrombin could not bring some useful information as well, by comparing basal condition with fibrinogen component (see for instance PMID: 31878831 DOI: 10.1080/09537104.2019.1704713) What are Haema system and CFMS LEPU (Table 1)?
How many clinical trials and relevant meta-analyses 
(Table 1) could be retrived?
Several studies were identified that compared the use of TEG vs. 
L.116 LTA to monitor platelet function: I don’t see how there could be randomized studies to do such a comparison between PFT.
A short but well-informative insight into VET and speciually technical differnecs betewwen former TEG devices and TEG®6s, and about PalteletMapping, is required. Some hints of the precision and reliability of TEG could have been incorporated in the text, and also about differences between the two TEG devices, which are not trivial. It should be made clear to what extent the different TEG devices yield similar – interchangeable results. Is there any study about TEG over time during APT? See for instance Campo G, Parrinello G, Ferraresi P, et al. Prospective evaluation of on-clopidogrel platelet reactivity over time in patients treated with percutaneous coronary intervention relationship with gene polymorphisms and clincal outcome. J Am Coll Cardiol 2011;57:2474–83. In other words, is there any evidence that testing at a unique time-point reflects platelet inhibition during chronic, long-term treatment?
Of note it is argued by some that expressing the changes of mechanical properties as amplitudes and not elasticity underestimate the platelet component. Please comment.
I do not agree on the phrasing ‘percentage platelet aggregation rate’ in Table 2: this is platelet contribution to clot formation, as assessed with its mechanical properties – to what extent platelet-to-platelet aggregation occurs and contributes to the signal is not clear to me at all – please provide facts about that important point. I do not understand P. 11 “These platelet reactivity measures, ADP.MA and AA.MA, were shown to be superior to platelet inhibition rate such as AA.%inhibition and ADP.%inhibition in assessing thrombotic risk... »
I would avoid the term ‘resistance’. Stick to hyporesponsiveness / HTPR.
L.114: this is not validation - analytical validation? Clinical validation?
L.221: more efficient than what?
Table 3 and Fig.3 are useful, but tables summarizing the studies should be more systematic, by indicating the country where the study was conducted, and above all the design and the specific TEG device used. In addition, timing of blood collection and of drug intake should be mentioned. The corresponding text is rather long.
There does not seem to be unequivocal evidence that using TEG-PM improves clinical outcome by guiding APT: choice of the drug, and dosing. In that respect it reads LL.62-63 “Assessment of patient response to DAPT is important as responsiveness can differ between patients »; the fact that such differences do exist does not suffice to make it important to resort to PFT in clinical practice: it would be really important (= practice-changing) if, and only if, there would be a validated clinical impact. 
How many of the examples given just below in the manuscript rely on VET?
« The most robust study identified in this regard was the CREATIVE trial; this RCT provides strong evidence that escalation of DAPT based on a low TEG-determined platelet inhibition rate improves clinical outcomes over an 18-month follow-up period, without compromising safety »: so there would be just one single such study? [45]
P.12 L.90: I strongly disagree that the identified studies are of high methodological quality – can’t be so with what the authors themselves identify and list just after in the paragraph.
I do not understand the structuration with first ‘Utility of thromboelastography to guide individualized treatment’ 
followed ‘…with guiding escalation of antiplatelet therapy’. Regarding the first, there is a mix of studies with different designs and objectives. Regarding the second, the more so since we can read below in the manuscript P.14 LL.205 et sq. that “…recent European guidelines suggesting that whole blood platelet function testing may be used to guide de-escalation of therapy but not escalation”. Ref #53 is a good one but inappropriate here: original papers to be cited = 2017 ESC focused update on DAPT…doi: 10.1093/eurheartj/ehx419.PMID: 28886622 and doi: 10.1016/j.jcin.2019.03.034. Epub 2019 Jun 12.PMID: 31202949
Please be more cautious and carefully quote those guidelines:
“…. neither platelet function testing nor genetic testing can be recommended for tailoring DAPT. It may be considered in specific situations (e.g. patients suffering from recurrent adverse events) if the results may change the treatment strategy.”
“ In selective scenarios, PFT and genotyping may be used as optional tools for guiding treatment. »
I suggest having a look also at doi: 10.1016/j.jacc.2018.09.057 PMID: 30522654
A table would be useful here as well. On page 11, “Following DAPT adjustment guided by 
L.86 TEG… » 
Please specify. Importantly, this is a retrospective study.
Moreover the second contains only two studies on escalation, and only one RCT.
Are they no studies on de-escalation?
Again ‘utility’ is just potential.
It is far form clear to me which parameters of TEG-PM would be the more informative about platelet inhibition and clinical outcomes.
…guide individualized treatment in several 
L.49 studies from East Asia: I do not analyse those studies as investigating APT guiding properly – just descriptive, associations.
‘East-Asian studies’ seems to mean studies conducted in continental China only.
Non-Asian studies should be summarized: is there any with prospective assessment of the clinical impact of using TEG to personalise APT? The text, tables and the figure 3 do not explicitly delineate what was conducted in East Asia.
The place of VET / TEG among the available tools for PFT is left unaddressed.
The abstract is poorly informative; it does not contain any numerical result, and even nothing about the number of studies and patients.
The authors, who are employees of the TEG company, are too optimistic in their statements about the usefulness of TEG to guide APT.
For instance it reads P.13 “Since TEG has 
L.194 been widely validated vs. LTA in both Western and East Asian populations … » again what does ‘validated’ mean? when is ‘widely’ a fully appropriate?

Elsewhere “TEG can provide rapid near-patient assessment of platelet function and provide clinicians with a valuable clinical assessment of patients’ response to anti-platelet therapy, which can, therefore, guide clinical decision-making »
- this is not a CLINICAL assessment!
- “can… guide” to be changed to which therefore might be a good candidate tool to guide…
P.13 divided into two groups: normal and hyperbolic 
L.145 types (including normal and hyperbolic), …: Please check the wording and the structure
Western: to be written with a capital letter everywhere in the text.
Please check the structure of the following sentence P.14: Currently, recommendations for the use of near-patient assessments, such as throm- 
L.205 -boelastography, to tailor antiplatelet use vary in guidelines, with recent European guidelines suggesting that …

LL. 34-35: “…antiplatelets … to dissolve clots »
antiplatelet gents
clots = thrombi
I am unsure that APA can ‘dissolve’ thrombi (= thrombolysis), perhaps anti-IIb/IIIa ?
LL.38-39: Monitoring of hemostasis in relation to ACS/PCI is crucial to optimize coagulation pre-PCI to reduce risk of bleeding and thrombosis … but PFT do not seem to be implemented in most centres!
P.2 L.57: “holistic overview of the patient’s hemostatic status” 

VET does not explore platelet adhesion, aggregation, secretion.
In addition, to what extent platelet procoagulant phospholipid exposure is explored is to my knowledge not fully known.
L.80 ADP and arachidonic acid (AA) assays 
does not mean very much: most if not all PFT aiming at quantifying antiplatelet effects are based on those platelet activators
P.13 L.188: check the appropriateness of ref #23 here
Ref#3 is specifically on bivalirudin and ACT, thus does not match with the sentence to which it is attached.
Bibliography on recent good reviews on PFT and APT not updated
Among others, check
doi: 10.1016/S0140-6736(21)00533-X.PMID: 33865495
doi: 10.2174/1570161117666190513105859.PMID: 31092181
doi: 10.3390/jcm9010189.PMID: 32284512
Is this reference used?
2020 Asian Pacific Society of Cardiology Consensus Recommendations on the Use of P2Y12 Receptor Antagonists in the Asia-Pacific Region.
Tan JW, Chew DP, Abdul Kader MAS, Ako J, Bahl VK, Chan M, Park KW, Chandra P, Hsieh IC, Huan DQ, Johar S, Juzar DA, Kim BK, Lee CW, Lee MK, Li YH, Almahmeed W, Sison EO, Tan D, Wang YC, Yeh SJ, Montalescot G.Eur Cardiol. 2021 Mar 2;16:e02. doi: 10.15420/ecr.2020.40. eCollection 2021 Feb.PMID: 33708263
Author Response
This is a systematic review in the field of (D)APT, that deals with two significant issues
- potential ethnical differences in the antiplatelet effects
- potential usefulness of VET / TEG
The issue of escalation and de-escalation is important indeed (see for instance doi: 10.1016/j.amjcard.2020.12.020.PMID: 33706987)
Regarding the first point, the authors should better delineate the clinical impact: Are ethnic differences potentially clinically relevant? How many patients of East-Asian ascent were included in the published clinical studies? North-Am based studies vs. Asian studies. What were the outcomes according to ethnical background? Might traditional Chinese medicines if used ‘on top of’ APT have played a role? Were their uses controlled for? And what about dietary influences?
Response: We have added a sentence into the introduction regarding additional factors, including diet and traditional medicine use, that may have affected this East Asian population. Regarding the first point, this review did not seek to compare an East Asian population with a Western population, only to explore the use of the TEG® device in East Asian study centres. All studies identified through the screening from all four databases were in East Asia, as this was one of the inclusion criteria. We, therefore, did not explore outcomes by background. While we address the possibility of possible differences due to genetic factors, this study is not intended to assess these differences or compare populations. Future comparison studies may be warranted to address the population differences based on this study overall findings.
Is there any interventional cardiologist working in China among the authors? Author Contributions: J.H., J.D. and O.X. contributed to the design of the study; J.D. and O.X. contributed to the screening of papers and data extraction; all authors contributed to the interpretation of the data, writing of the manuscript and reviewed/approved the final version for submission. What was the role of Tang Yi-Da then?
Response: We have clarified in the Author Contributions that Professor Tang Yi-Da, who is an interventional cardiologist at Peking University, was involved in the data interpretation, manuscript writing and reviewing, and approved the final version for submission.
L.256 Conflicts of Interest: J.H., J.D. and O.X. are employees of Haemonetics. The remaining authors declare no competing interests: just one remaining author! Namely Tang Yi-Da.
Response: We have clarified that Professor Tang Yi-Da had no competing interests.
Importantly, the authors conclude LL173 et sq. that “studies from this region support the comparability of TEG to traditional laboratory parameters, i.e., LTA, as well as other whole blood near-patient platelet function testing assays (VerifyNow® ), in the context of PCI/AMI. Studies from East Asia have also shown TEG to be effective in guiding hemostasis management in relation to individualized treatment, and that TEG platelet FUNCTION parameters are associated with cardiovascular outcomes”. This is not fully supported by the literature: first one, to some extent only – PFT using different read-outs can’t be easily compared – classification of patients according to the extent of platelet inhibition; second: I strongly disagree - just one single RCT?; third: I agree – but should be in the second place.
Response: We have modified the wording of the first paragraph of the discussion in line with the reviewer’s comments.
Literature search and grading were well conducted and described. Why was a meta-analysis not possible? Of note, English but also Chinese language databases were included. For the latter, just one author could analyse them, which is a study limitation acknowledged by the authors.
Response: This study was not designed around a specific research question that a meta-analyses would have been formulated to answer. Instead, it aimed to give an overview of literature describing the use of thromboelastography in a cardiology setting in the Asian population. Future meta-analyses may be warranted to address specific research questions based on these overall findings.
PRISMA guidelines were used. Tables are welcome: S1: Search terms; S2: SIGN grading but Table S2 should mention the language of the retrieved studies. I have doubts about the evaluation of methodological quality of identified studies. For instance P.13 it reads “…deemed to be generally of low risk of bias. » The authors do not provide convincing evidence for such an appreciation.
Response: Table S2 is divided into two sections, the first lists the manuscripts from the Western databases which were all written in English, the second lists the manuscripts from the Chinese databases, which were all available in Mandarin Chinese. We have included more detail in the table headings to indicate which language the studies were written in as well as which database. A bias analysis was carried out which found the majority of studies, including those discussed at length in the sections, were of SIGN grade 1 or 2 (see supplementary information Table 2). We have modified the statements regarding risk of bias to address this. As the reviewer points out in a later comment, it would not be possible to design a SIGN Grade 1 study comparing two platelet function tests as the treating physician could not be blinded to the information from the devices used to make the clinical judgement.
Studies in the setting of AVR were few and the authors decided not to incorporate them into their review: please tell us how many they were.
Response: Following application of all inclusion/exclusion criteria, three studies were included that were in the setting of aortic valve replacement. Search terms were used for valve replacement as TEG® PlateletMapping® has potential utility in this setting. However, due to the limited number (three) and scope of these included studies, we decided to focus the review on the PCI setting. As suggested, we have added the number of valve replacement studies that met the inclusion/exclusion criteria into the results section.
The retrieved studies are with TEG only by design of the systematic review; hence ‘thromboelastography’ should be ‘viscoelastometric testing (VET) with TEG® device’. Admittedly TEG® with PM (PlateletMapping®) is the most elaborated approach to study with VET the effect of APT on platelets – it is a complicated one, but the authors could have discussed why other VET devices with the use of an inhibitor of the platelet component and platelets stimulated by endogenous thrombin could not bring some useful information as well, by comparing basal condition with fibrinogen component (see for instance PMID: 31878831 DOI: 10.1080/09537104.2019.1704713) What are Haema system and CFMS LEPU (Table 1 )?
Response: Without the use of the PlateletMapping cartridge, the TEG® device can assess the platelet contribution to clot strength. However, this only measures the effect of thrombin, rather than the more sensitive measures of the response to ADP and AA measured by the PlateletMapping® cartridge. Due to this increased sensitivity, it is not an equal comparison between the TEG® Plateletmapping® device and other VETs only using the thrombin component of platelet function. We, therefore, excluded all other devices without a specific platelet function that relates to pharmacological inhibition of the ADP and AA pathways. The Haema TX from Medcaptain is a generic whole blood thromboelastography system (https://www.medicalexpo.com/prod/medcaptain-medical-technology/product-118704-976735.html), as is the CFMS from LEPU Medical (https://en.lepumedical.com/products/cfms-lepu-8880/).
How many clinical trials and relevant meta-analyses (Table 1) could be retrieved?
Response: We have added the number of RCTs, which were listed in Table S2; no meta-analyses were identified.
Several studies were identified that compared the use of TEG vs. LTA to monitor platelet function: I don’t see how there could be randomized studies to do such a comparison between PFT.
Response: Most studies that reported methods comparisons were non-randomized prospective studies. In RCTs that included comparison of methods, randomization was between treatment regimens, rather than PFT method.
A short but well-informative insight into VET and speciually technical differnecs betewwen former TEG devices and TEG®6s, and about PalteletMapping, is required. Some hints of the precision and reliability of TEG could have been incorporated in the text, and also about differences between the two TEG devices, which are not trivial. It should be made clear to what extent the different TEG devices yield similar – interchangeable results. Is there any study about TEG over time during APT? See for instance Campo G, Parrinello G, Ferraresi P, et al. Prospective evaluation of on-clopidogrel platelet reactivity over time in patients treated with percutaneous coronary intervention relationship with gene polymorphisms and clincal outcome. J Am Coll Cardiol 2011;57:2474–83. In other words, is there any evidence that testing at a unique time-point reflects platelet inhibition during chronic, long-term treatment?
Response: Previous studies have compared the TEG®5000 and the TEG®6s devices, including in a cardiac surgery clinical setting. We have added these references, along with a brief sentence explaining, to the discussion section. Regarding the time-point of TEG® assessments, we agree with the reviewer that this is an important topic; however, we do not feel there is currently enough evidence regarding timing of assay treatment to address it in depth. We have added the assay timing into Table 3; however, we note that this was not a study question in any of the literature extracted. We have also added a sentence to the discussion to acknowledge that further evidence is required, particularly in relation to the timing of thromboelastography monitoring.
Of note it is argued by some that expressing the changes of mechanical properties as amplitudes and not elasticity underestimate the platelet component. Please comment. I do not agree on the phrasing ‘percentage platelet aggregation rate’ in Table 2: this is platelet contribution to clot formation, as assessed with its mechanical properties – to what extent platelet-to-platelet aggregation occurs and contributes to the signal is not clear to me at all – please provide facts about that important point. I do not understand P. 11 “These platelet reactivity measures, ADP.MA and AA.MA, were shown to be superior to platelet inhibition rate such as AA.%inhibition and ADP.%inhibition in assessing thrombotic risk... »
Response: The ability of platelets to be activated by the specific pathway/receptor is captured by the strength of the clot that is formed when the activated platelets attach to the fibrin mesh. The platelet function can, therefore, be read either with the maximum clot amplitude from the ADP or AA activated assay, or using the calculation to provide % inhibition or % aggregation. We have reviewed the wording around the TEG® device within the manuscript and do not feel a re-write in this case is warranted as a full description of the mechanism by which the TEG® device measures platelet function is out of the scope of this review. Regarding the parameters explored, and the terms used to refer to them, we have used the parameters recorded by the studies to report the platelet component of thrombosis. The percentage platelet aggregation rate is a term approved/cleared by a number of different regulatory bodies and is, therefore, used for consistency across publications.
I would avoid the term ‘resistance’. Stick to hyporesponsiveness / HTPR.
Response: We have changed this wording where it appears as per the reviewer’s suggestion.
L.114: this is not validation - analytical validation? Clinical validation?
Response: We thank the reviewer for raising this question; the data presented are methods comparisons in the clinical setting (ACS patients), and, therefore, feel that this represents clinical validation.
L.221: more efficient than what?
Response: More efficient compared with genetic testing. We have altered the text of the sentence to clarify this point.
Table 3 and Fig.3 are useful, but tables summarizing the studies should be more systematic, by indicating the country where the study was conducted, and above all the design and the specific TEG device used. In addition, timing of blood collection and of drug intake should be mentioned. The corresponding text is rather long.
Response: As per the reviewer’s suggestion, we have added the TEG® device used, and the timing of the blood collection and drug intake. All studies in this table were conducted in China, and we have added this into the table heading, along with a note in the footer about which studies were from the Western and which from the Chinese databases. We have not included the study design in this table, as this information is included for each study in Table S2.
There does not seem to be unequivocal evidence that using TEG-PM improves clinical outcome by guiding APT: choice of the drug, and dosing. In that respect it reads LL.62-63 “Assessment of patient response to DAPT is important as responsiveness can differ between patients »; the fact that such differences do exist does not suffice to make it important to resort to PFT in clinical practice: it would be really important (= practice-changing) if, and only if, there would be a validated clinical impact. How many of the examples given just below in the manuscript rely on VET?
Response: While the use of platelet function tests are not yet routine in clinical practice, there are still a percentage of patients who experience adverse outcomes, either bleeding or thrombotic, on antiplatelet therapy. There is a growing body of evidence showing the importance of PFT in this clinical area and their potential to improve clinical outcomes. We have modified the sentence to clarify this point.
The most robust study identified in this regard was the CREATIVE trial; this RCT provides strong evidence that escalation of DAPT based on a low TEG-determined platelet inhibition rate improves clinical outcomes over an 18-month follow-up period, without compromising safety »: so there would be just one single such study? [45]
Response: The CREATIVE study was the only RCT identified that specifically used the TEG® device to escalate DAPT; however, there is a wider body of supportive evidence from prospective and observational trials showing that it relates to clinical outcomes and can distinguish between different intensities of antiplatelet treatment. There are also large-scale RCTs of different platelet function test technologies, such as the TROPICAL study on the use of Multiplate for de-escalation of DAPT, which support the overall use of platelet function tests for guiding therapy in patients on DAPT. Nevertheless, we acknowledge the reviewer’s point and have noted in the discussion that further evidence is required to support the specific use of TEG® in this manner.
P.12 L.90: I strongly disagree that the identified studies are of high methodological quality – can’t be so with what the authors themselves identify and list just after in the paragraph.
Response: We have removed the sentence in question, and instead left the list of the different types of studies.
I do not understand the structuration with first ‘Utility of thromboelastography to guide individualized treatment’ 
followed ‘…with guiding escalation of antiplatelet therapy’. Regarding the first, there is a mix of studies with different designs and objectives. Regarding the second, the more so since we can read below in the manuscript P.14 LL.205 et sq. that “…recent European guidelines suggesting that whole blood platelet function testing may be used to guide de-escalation of therapy but not escalation”. Ref #53 is a good one but inappropriate here: original papers to be cited = 2017 ESC focused update on DAPT…doi: 10.1093/eurheartj/ehx419.PMID: 28886622 and doi: 10.1016/j.jcin.2019.03.034. Epub 2019 Jun 12.PMID: 31202949
Response: We have clarified the wording of the headings and introductory sentences of these sections to make it clear that the first section is a collection of studies exploring the use of the TEG® device to measure patient response to therapy and optimize antiplatelet use, while the second is specifically studies regarding escalation of therapy. This second section is of particular interest given that, as the reviewer notes, European guidelines do not focus on this particular use of platelet function tests, instead focusing on de-escalation of therapy. Hence, we have kept it as a separate section in the manuscript. We have updated the guideline references as suggested by the reviewer.
Please be more cautious and carefully quote those guidelines:
“…. neither platelet function testing nor genetic testing can be recommended for tailoring DAPT. It may be considered in specific situations (e.g. patients suffering from recurrent adverse events) if the results may change the treatment strategy.”
“ In selective scenarios, PFT and genotyping may be used as optional tools for guiding treatment. »
I suggest having a look also at doi: 10.1016/j.jacc.2018.09.057 PMID: 30522654
Response: We thank the reviewer for highlighting these guideline statements, and have added them into the conclusion section as suggested.
A table would be useful here as well. On page 11, “Following DAPT adjustment guided by 
L.86 TEG… » 
Please specify. Importantly, this is a retrospective study.
Response: As per the earlier suggestion, we have updated Table 3 to include details of timing of treatment and TEG® device measurements. The study on page 11 is noted as a retrospective case control study at the beginning of the paragraph; however, we have made sure each study design is noted as studies are discussed.
Moreover the second contains only two studies on escalation, and only one RCT.
Are they no studies on de-escalation?
Response: In the studies identified through this meta-analysis no studies on de-escalation were identified. This is likely to be due to the different aspects of the platelet function tests being explored in East Asia, where there seems to be more focus on using the TEG® device to optimize efficacy of treatment, rather than mitigating risk. The expanded section in the discussion on the guidelines now draws out this point further.
Again ‘utility’ is just potential.
Response: Wording has been changed in the summary paragraph to make clear this is a potential utility of the TEG® device rather than an established common use.
It is far form clear to me which parameters of TEG-PM would be the more informative about platelet inhibition and clinical outcomes.
Response: This manuscript does not set out to determine the exact parameters of the TEG-PM® device that are most optimal for clinical use. Rather, it is an exploration of studies from East Asia, looking at the different ways the TEG® device is used in this region. As stated in the discussion section, further studies would be needed to determine the ideal TEG® assay parameters for clinical use.
…guide individualized treatment in several 
L.49 studies from East Asia: I do not analyse those studies as investigating APT guiding properly – just descriptive, associations.
Response: We have altered the statement to make it clearer what the studies are assessing, and again to stress that the use of the TEG® device to guide individualized treatment is a potential utility.
‘East-Asian studies’ seems to mean studies conducted in continental China only.
Response: Our original search (as detailed in the methods section) was performed in PubMed and three Chinese databases. The search sought to capture papers from East Asian countries, namely China, Taiwan, Japan and Korea. However, nearly all papers that met the pre-specified screening criteria were from China.
Non-Asian studies should be summarized: is there any with prospective assessment of the clinical impact of using TEG to personalise APT? The text, tables and the figure 3 do not explicitly delineate what was conducted in East Asia.
Response: We have updated the title of Figure 3 to make it clear these studies were all carried out in China. There are already a number of systematic and narrative reviewers on the use of thromboelastography in non-Asian countries, and, therefore, this review was focused on the results from East Asian studies.
The place of VET / TEG among the available tools for PFT is left unaddressed.
Response: As per the comment above, the aim of this study was not to reach a decision on the value of the TEG® device compared with other platelet function tests. As the reviewer has pointed out, the evidence base is not strong enough for such a conclusion to be made, with the lack of large numbers of rigorous RCTs. Instead this review aimed to collate data on the use of the TEG® device in East Asia to share clinical practice and guide clinicians for potential areas of future research.
The abstract is poorly informative; it does not contain any numerical result, and even nothing about the number of studies and patients.
Response: We have updated the abstract to include more details of the numbers of studies identified. As the review is exploratory in nature and not a meta-analysis, we are limited as to the data that can be presented; we, therefore, aimed to provide an overview of our findings.
The authors, who are employees of the TEG company, are too optimistic in their statements about the usefulness of TEG to guide APT.
For instance, it reads P.13 “Since TEG has 
L.194 been widely validated vs. LTA in both Western and East Asian populations … » again what does ‘validated’ mean? when is ‘widely’ a fully appropriate?
Response: We have clarified in this instance that the TEG® device has been found to correlate with the gold-standard LTA for identifying platelet response.
Elsewhere “TEG can provide rapid near-patient assessment of platelet function and provide clinicians with a valuable clinical assessment of patients’ response to anti-platelet therapy, which can, therefore, guide clinical decision-making »
- this is not a CLINICAL assessment!
- “can… guide” to be changed to which therefore might be a good candidate tool to guide…
Response: We have modified the wording as suggested by the reviewer.
P.13 divided into two groups: normal and hyperbolic 
L.145 types (including normal and hyperbolic), …: Please check the wording and the structure
Western: to be written with a capital letter everywhere in the text.
Response: We have made both edits as suggested.
Please check the structure of the following sentence P.14: Currently, recommendations for the use of near-patient assessments, such as throm- 
L.205 -boelastography, to tailor antiplatelet use vary in guidelines, with recent European guidelines suggesting that …

Response: This sentence has now been modified in response to a previous comment
- 34-35: “…antiplatelets … to dissolve clots »
antiplatelet gents
clots = thrombi
I am unsure that APA can ‘dissolve’ thrombi (= thrombolysis), perhaps anti-IIb/IIIa ?
Response: The clause ‘to dissolve clots’ in this sentence is in reference to fibrinolytics, which we list at the beginning of the sentence. We have slightly altered the wording in this sentence in response to the comment.
LL.38-39: Monitoring of hemostasis in relation to ACS/PCI is crucial to optimize coagulation pre-PCI to reduce risk of bleeding and thrombosis … but PFT do not seem to be implemented in most centres!
Response: We have amended these sentences to separate out what are standard haemostatic tests (e.g., platelet count and APTT) and what are additional tests that can be employed, i.e., PFT.
P.2 L.57: “holistic overview of the patient’s hemostatic status” 

VET does not explore platelet adhesion, aggregation, secretion.
In addition, to what extent platelet procoagulant phospholipid exposure is explored is to my knowledge not fully known.
Response: We take the reviewer’s point that the TEG device does not assess every aspect of platelet activity; we have, therefore, changed the wording to “global haemostasis assay.”
L.80 ADP and arachidonic acid (AA) assays 
does not mean very much: most if not all PFT aiming at quantifying antiplatelet effects are based on those platelet activators
Response: This line in the introduction is explaining which pathways are used for testing platelet activity, as the reviewer states these are the pathways analysed by all PFTs. This is not intended to imply that analysing these pathways is exclusive to the TEG® device; in fact, the TEG® device is not mentioned in this paragraph, which focuses on an overview of genetic DAPT resistance and how this is measured in patients.
P.13 L.188: check the appropriateness of ref #23 here
Ref#3 is specifically on bivalirudin and ACT, thus does not match with the sentence to which it is attached.
Response: We have checked through these references as suggested and changed to more appropriate references.
Bibliography on recent good reviews on PFT and APT not updated
Among others, check
doi: 10.1016/S0140-6736(21)00533-X.PMID: 33865495
doi: 10.2174/1570161117666190513105859.PMID: 31092181
doi: 10.3390/jcm9010189.PMID: 32284512
Is this reference used?
2020 Asian Pacific Society of Cardiology Consensus Recommendations on the Use of P2Y12 Receptor Antagonists in the Asia-Pacific Region.
Tan JW, Chew DP, Abdul Kader MAS, Ako J, Bahl VK, Chan M, Park KW, Chandra P, Hsieh IC, Huan DQ, Johar S, Juzar DA, Kim BK, Lee CW, Lee MK, Li YH, Almahmeed W, Sison EO, Tan D, Wang YC, Yeh SJ, Montalescot G.Eur Cardiol. 2021 Mar 2;16:e02. doi: 10.15420/ecr.2020.40. eCollection 2021 Feb.PMID: 33708263
Response: Many thanks to the reviewer for highlighting these additional references. We have added into the manuscript where appropriate.

Reviewer 3 Report
The authors of “The use of thromboelastography in percutaneous coronary intervention and acute coronary syndrome in East Asia: A systematic literature review” have done an extensive and thorough systematic literature review regarding the use of TEG in PCI. They have compared it with other modalities, most notably light transmission aggregometry (LTA) and found it to be comparable and logistically more useful as it is a point of care test. The high prevalence of CYP2C19 poor metabolizers in China make it even more important in customizing DAPT therapy. Often, physicians use clopidogrel without determining platelet reactivity.
The authors have reported in detail the results of their exhaustive literature review in a rich narrative as well as with tables, references, and an algorithmic flow chart. The only question I have for the authors is whether it would be important to know when the TEG 5000 or the TEG 6s was used. Is there the potential with the TEG 6s ADP and AA assay cartridge that there would be more accuracy and less inter- and intra- observer variability than with the TEG 5000?
Author Response
The authors of “The use of thromboelastography in percutaneous coronary intervention and acute coronary syndrome in East Asia: A systematic literature review” have done an extensive and thorough systematic literature review regarding the use of TEG in PCI. They have compared it with other modalities, most notably light transmission aggregometry (LTA) and found it to be comparable and logistically more useful as it is a point of care test. The high prevalence of CYP2C19 poor metabolizers in China make it even more important in customizing DAPT therapy. Often, physicians use clopidogrel without determining platelet reactivity.
The authors have reported in detail the results of their exhaustive literature review in a rich narrative as well as with tables, references, and an algorithmic flow chart. The only question I have for the authors is whether it would be important to know when the TEG 5000 or the TEG 6s was used. Is there the potential with the TEG 6s ADP and AA assay cartridge that there would be more accuracy and less inter- and intra- observer variability than with the TEG 5000?
Response: Many thanks to the reviewer for their time and comments on the manuscript. We have added a column into Table 3, which details which device is used in each study, and also included a brief referenced sentence in the introduction on the correlation between the two TEG® devices.
Round 2
Reviewer 2 Report
the last sentence of the abstract must be modified"... can guide treatment in L.27 these patients. " --> could - further studies are required
P.12 L.52: this crucial point is much better phrased " This could potentially be used to guide individualized treatment by adjusting therapy depending on patient response."
#1
We, therefore, did not explore outcomes by background. While we address the possibility of possible differences due to genetic factors, this study is not intended to assess these differences or compare populations. Future comparison studies may be warranted to address the population differences based on this study overall findings.
= a study limitation that should be mentioned as such.
#9 Several studies were identified that compared the use of TEG vs. LTA to monitor platelet function: I don’t see how there could be randomized studies to do such a comparison between PFT.
Response: Most studies that reported methods comparisons were non-randomized prospective studies. In RCTs that included comparison of methods, randomization was between treatment regimens, rather than PFT method.
Thanks for the response - point to be clarified in the text.
#10 mostly unaddressed
Notably there is a misunderstanding about study about TEG over time during APT
I was referring to longitudinal studies.
|
A short but well-informative insight into VET and specifically technical differences between former TEG devices and TEG®6s, and about PlateletMapping, is required. Some hints of the precision and reliability of TEG could have been incorporated in the text, and also about differences between the two TEG devices, which are not trivial. It should be made clear to what extent the different TEG |
|
devices yield similar – interchangeable results. Is there any study about TEG over time during APT? See for instance Campo G, Parrinello G, Ferraresi P, et al. Prospective evaluation of on-clopidogrel platelet reactivity over time in patients treated with percutaneous coronary intervention relationship with gene polymorphisms and clincal outcome. J Am Coll Cardiol 2011;57:2474–83. In other words, is there any evidence that testing at a unique time-point reflects platelet inhibition during chronic, long-term treatment? Response: Previous studies have compared the TEG®5000 and the TEG®6s devices, including in a cardiac surgery clinical setting. We have added these references, along with a brief sentence explaining, to the discussion section. Regarding the time-point of TEG® assessments, we agree with the reviewer that this is an important topic; however, we do not feel there is currently enough evidence regarding timing of assay treatment to address it in depth. We have added the assay timing into Table 3; however, we note that this was not a study question in any of the literature extracted. We have also added a sentence to the discussion to acknowledge that further evidence is required, particularly in relation to the timing of thromboelastography monitoring. |
#11 largely unaddressed
In particular
I do not understand at all "platelet component of thrombosis" - thrombosis is by essence an in vivo phenomenon
|
ADP.MA |
|
and AA.MA, AA.%inhibition and ADP.%inhibition |
should be made more explicit
above all, the point about elasticity is not dealt with.
#13
|
L.114: this is not validation - analytical validation? Clinical validation? |
|
comparisons in the clinical setting (ACS patients), and, therefore, feel that this represents clinical validation. |
--> observational clinical studies
#16 Please copy in your response the part of the text dealing with the point
|
There does not seem to be unequivocal evidence that using TEG-PM improves clinical outcome by guiding APT: choice of the drug, and dosing. In that respect it reads LL.62-63 “Assessment of patient response to DAPT is important as responsiveness can differ between patients »; the fact |
|
that such differences do exist does not suffice to make it important to resort to PFT in clinical practice: it would be really important (= practice-changing) if, and only if, there would be a validated clinical impact. How many of the examples given just below in the manuscript rely on VET? Response: While the use of platelet function tests are not yet routine in clinical practice, there are still a percentage of patients who experience adverse outcomes, either bleeding or thrombotic, on antiplatelet therapy. There is a growing body of evidence showing the importance of PFT in this clinical area and their potential to improve clinical outcomes. We have modified the sentence to clarify this point. |
